# The relationship between psychological capital, patient's contempt, and professional identity among general practitioners during COVID-19 in Chongqing, China

Jingzhi Deng[1]◉, Yang Xu[1]◉, Qiaoya Li[2], Wen Yang[1], Huisheng Deng[1]*

1 Department of General Practice, The First Affiliated Hospital of Chongqing Medical University, Chongqing, People's Republic of China, 2 Department of General Practice, Yan'an Affiliated Hospital of Kunming Medical University, Kunming, People's Republic of China

◉ These authors contributed equally to this work.
* dhs700214@163.com

**Data Availability Statement:** All relevant data are within the paper and its Supporting Information files.

## Abstract

General practitioners are crucial in the primary healthcare system as well as for epidemic prevention and control. However, few researchers have examined their professional identity. This study investigated the current status of the professional identity of general practitioners in Chongqing, China and explored the effects of psychological capital and patient's contempt on their professional identity. From December 2021 to January 2022, randomized cluster sampling was used to conduct a cross-sectional online self-assessment questionnaire survey among general practitioners in Chongqing. In total, 2,180 general practitioners working for more than one year were selected. General practitioners' sense of professional identity, mental health, and sense of patients' disrespect were measured using the Professional Identity Scale, Psychological Capital Questionnaire, and Patient's Contempt Questionnaire. Sociodemographic characteristics were also collected. A multiple linear regression model was used to analyze the association between professional identity, psychological capital, and patient's contempt. The average score for professional identity among general practitioners was 53.59 (SD = 6.42). The scores for self-efficacy, hope, resilience, and optimism (subscales of psychological capital) were 26.87 (SD = 5.70), 26.47 (SD = 5.74), 26.97 (SD = 5.55), and 26.86 (SD = 5.59), respectively. The score for perceived contempt was 34.19 (SD = 7.59). An average monthly income greater than CNY 8,000 (β = 1.018, p < 0.001), work tenure of more than 15 years (β = 0.440, p = 0.001), hope (β = 0.249, p < 0.001) and a higher optimism score (β = 0.333, p < 0.001) were positively correlated with professional identity. Having a bachelor's degree and above (β = -0.720, p = 0.014), an administrative role (β = -1.456, p < 0.001), self-efficacy (β = -0.122, p < 0.001), and higher patient's contempt (β = -0.103, p < 0.001) were negatively associated with professional identity. General practitioners in Chongqing demonstrated high professional identity and a strong psychological status during the COVID-19 pandemic. Psychological capital and patient's contempt were associated with professional identity. To improve general

**Funding:** HSD 2020ZLXM003 Joint project of Chongqing Health Commission and Science and Technology Bureau http://wsjkw.cq.gov.cn/ http://kjj.cq.gov.cn/ The funders had a role in data collection of the manuscript.

**Competing interests:** The authors have declared that no competing interests exist.

practitioners' professional identity, stakeholders should promote practitioners' mental health and physician–patient relationships in China.

## Introduction

General practitioners (GPs), also known as family physicians, are the main providers of primary healthcare services and the "gatekeepers" of human health [1, 2]. Since the outbreak of COVID-19 in Wuhan, China, at the end of 2019, GPs have actively responded to the country's call and played an indispensable role in prevention and control activities, such as the delivery of home care for community residents, screening of patients with fever, and preventive triage [3–5]. Nonetheless, many countries, such as the United States, Australia, and South Korea, have long faced a shortage of GPs [6–8]. As the largest developing country in the world, the development of a general practice system in China occurred relatively late. Moreover, because of the increasing demand for primary healthcare services by the Chinese population, the number of GPs is becoming relatively insufficient [9, 10].

The social identity of GPs also tends to be lower than that of specialist physicians. Owing to their low social status, GPs are often easily ignored and not regarded as important in society, which leads to a higher turnover rate among such professionals [1, 11]. GPs in China have been in a high-intensity work environment for a considerable period of time, leading them to experience both physical and psychological fatigue [12, 13]. If this persists, this situation is likely to lead to a decline in GPs' perception of and emotional attachment to their own profession, thus affecting their professional identity.

Professional identity refers to one's view of the goal, social value, and other related factors of an occupation, which tends to be consistent with the social evaluation and expectation of that occupation [14, 15]. This concept tends to be indirectly reflected in students' transition to becoming physicians and managing the related daily life struggles [16]. Researchers have shown that professional identity impacts GPs' work engagement and career progress by affecting their sense of professional mission [17]. Greater professional identity can improve work quality and is conducive to patients' recovery from disease. However, there are relatively few investigations of the professional identity of GPs in China. According to a survey of the professional identity of GPs in China before the COVID-19 pandemic, the professional identity score of 3,236 GPs averaged at 51.23 (out of a maximum score of 65); further, because of the regional economic development imbalance in China, the professional identity of GPs in Central China was significantly higher than that of GPs in other regions, and the professional identity of GPs with high income (vs. low income) and long work experience (vs. short work experience) was higher. Research also shows that GPs outside China identify more with their profession; this may be because the general practice system has been established for longer and the social status of GPs is higher in countries other than China [18].

Psychological capital is a factor that has been shown to positively influence professional identity; it is defined as positive psychological states related to the process of individual development [19–21] and encompasses the concepts of self-efficacy, optimism, resilience, and hope.

GPs play the important role of ensuring citizen health management, implying that GPs often need to treat patients. However, owing to knowledge inequality between physicians and patients, GPs may often not be understood or respected by patients and even experience workplace violence. A mental health survey conducted with a sample of primary care physicians in China showed that 31.7% were facing mental health problems and depression [22]. Meanwhile, researchers have shown that positive psychological capital among GPs can alleviate

negative emotional experiences in the work environment and help these professionals deal with their negative emotions more quickly [23, 24]. Self-efficacy refers to one's confidence in one's own competency to complete a task, face challenges, and strive for success. Optimism refers to a person having a positive attribution style and attitude toward both the present and the future. Resilience is defined as the ability to quickly recover from adversity, setbacks, and failures, and even positively transform and grow after their occurrence. Hope is a state of positive motivation to achieve a predetermined goal through various means [25].

The emotional experiences of GPs with their patients during the COVID-19 pandemic also warrant attention. A harmonious physician–patient relationship is one of the foundational factors of GPs' professional identity [26]. However, contradictions between GPs and patients have long been a problem in Chinese society [27, 28]. First, owing to the imperfect Chinese primary healthcare system, patients in the country often face difficulties in seeing a medical practitioner. Second, with the development of the Internet, patients sometimes prefer to seek help from the Internet instead of trusting medical practitioners [29]. Moreover, medical practitioners are often worried about patients showing aggressive and threatening attitudes that endanger practitioner safety, leading to ongoing deterioration of the physician–patient relationship.

For the purposes of this study, the aforementioned behaviors of patients that endanger GPs' safety during physician–patient interactions will be referred to as "patient's contempt"; examples are GPs being disrespected or distrusted and patients and their families showing non-cooperation and contempt during physician–patient interactions. We believe that patients' contempt can indirectly reflect the contradiction between physicians and patients and may affect the professional identity of GPs. Although the physician–patient relationship has improved in China since the outbreak of COVID-19, it continues to be an urgent problem that is unaddressed countrywide.

In summary, professional identity is extremely important in the quality of GPs' work; however, despite this importance, few researchers have explored the professional identity of these healthcare professionals. This study investigated the current status and relationship between psychological capital, patient's contempt, and professional identity among GPs in Chongqing, China. We hypothesized that the professional identity of GPs would be improved during the COVID-19 pandemic, and greater psychological capital would promote the professional identity of these persons, while patients' contempt might lead to a reduction in the professional identity of GPs. Meanwhile, we also considered the factors that influence professional identity according to the general characteristics of GPs, such as age, gender, and professional title.

## Methods

### Study population and study design

A cross-sectional study of primary healthcare service centers in Chongqing, western China, was conducted from December 2021 to January 2022 using randomized cluster sampling. With the help of staff from the administrative department of Chongqing Municipal Health Commission, the investigation team selected primary health care service centers in 23 districts and counties, covering different levels of economic development and characteristics of urban and rural structure in Chongqing. The inclusion criteria were GPs working for more than one year in the primary health care service health centers in the selected districts and counties. Exclusion criteria were clinicians, nursing, laboratory, and other technical personnel who were not GPs in Chongqing medical and health institutions. This survey received a total of 2200 questionnaires from GPs. After excluding the data of 20 GPs owing to missing data or logic errors, a final sample of 2,180 valid questionnaires was obtained.

### Ethics approval

This study was approved by the institutional ethical review board of the first affiliated hospital of Chongqing Medical University. During the investigation, doctors clicked a link to view the informed consent form. If they clicked "agree," the questionnaire content was displayed; if they clicked "disagree," the investigation ended. Electronic informed consent was obtained from all participants.

### Measures

**Demographic characteristics.** General demographic characteristics in the questionnaire consisted of age, gender, educational background, marital status, contract status (GP wage, welfare paid by the Chinese government, whether the health institution could freely fire GPs), work tenure, income level, professional title, having an administrative role, monthly frequency of working overtime, working hours, workplace, and frequency of family visits and of data collection.

**Professional identity.** To assess professional identity, we used the 13-item Chinese version of the Professional Identity Scale from Zhao et al.'s and Wu's studies; Cronbach's alpha for this scale was 0.910 in their study [30, 31]. The scale includes statements such as "I consider my success as the success of health care workers" and "I care very much about other people's views of my career," and 11 other items (S1 File). Items are rated on a 5-point Likert scale ranging from 1–5 (strongly disagree–strongly agree), and item scores are summed to provide a composite score of professional identity (range: 13–65), with higher scores representing greater professional identity. Cronbach's alpha for the scale was 0.936 in the current study.

**Psychological capital.** Psychological capital was measured via the Psychological Capital Questionnaire (PCQ-24), which was developed by Luthans et al. and exhibited high internal consistency in their study [32]. The scale includes four dimensions, each of which is assessed via six items: self-efficacy (e.g., "I believe I can contribute to the discussion of the strategy of the company/community/organization"), hope (e.g., "There are many solutions to any problem"), resilience (e.g., "I usually take stress at work in stride"), and optimism ("I always look on the bright side of my work") (S1 File). Items are rated on a scale ranging from 1–6 (1, strongly disagree; 2, disagree; 3, somewhat disagree; 4, somewhat agree; 5, agree; 6, strongly agree), such that the higher the score, the greater the psychological capital. Cronbach's alpha was 0.977 in the current study.

**Patient's contempt.** Zhang used key person interviews, focus group discussions, and a literature review to sort, summarize, and determine the 10 common types of patient's contempt experienced by medical practitioners [33]. Next, they designed a questionnaire of physicians' perceptions of patient's contempt, which exhibited a Cronbach's α value of 0.935 in the original study. The questionnaire includes items such as "Patients and their families lack the ability of empathy, ignoring the fact that doctors also need to survive and should receive respect" and "In the process of doctor-patient disputes, patients and their families will choose to destroy and slander the social reputation of doctors (S1 File)." The questionnaire is scored on a 5-point scale (1, strongly disagree; 5, strongly agree), and total scores range from 10–50. The higher the score, the more serious the perception of patient's contempt by the medical practitioner. Cronbach's alpha for this scale was 0.914 in the current study.

### Data collection and quality control

The questionnaire in this study was designed based on a literature review, group discussions, and simulation interviews. To improve the quality of the questionnaire, GPs in Chongqing Medical University conducted a preliminary experiment using the instrument. Before the

survey, they communicated with the leaders of their department. After obtaining consent, the investigators issued a Questionnaire Star QR code to the GPs in the department (a web link for the online questionnaire, created by the Questionnaire Star software, was then disseminated to GPs through WeChat, the largest communication platform in China, with over one billion users). The participants provided demographic information and their views on professional identity, psychological capital, and patient's contempt.

## Statistical analysis

SPSS version 22.0 was used for data analysis. Qualitative data were described by the number of cases as a percentage (%), Quantitative data were described by the mean ± standard deviation (x ± s). Independent-samples t-tests and one-way analysis of variance were used to compare the professional identity scores of different demographic characteristics. Multiple linear regression was used to compare the psychological capital and patient's contempt according to different demographic characteristics. Pearson's correlation analysis and a multiple linear regression model were employed to analyze the association between psychological capital, patient's contempt, and professional identity. All tests were two-tailed, with a significance level of $p < 0.05$.

## Results

### Sample and professional identity scores

Of the 2,180 GPs, more than half were women (53.72%), and most were aged 30–49 years (67.1%). Furthermore, 64.95% of participants held a bachelor's degree, 80.92% were married, 86.65% had contracts at the preparation stage, and 94.55% had an average monthly income of CNY ≤ 8000. In addition, 53.07% held primary titles, 34.08% held intermediate titles, and 74.72% did not hold an administrative title. Finally, 97.57% of participants engaged in overtime work monthly, 682 (31.28%) had 15 years of tenure, and 71.97% of GPs in township hospitals undertook family visits and data collection. The scores for professional identity demonstrated statistically significant differences across age, gender, educational background, marital status, income level, professional title, management responsibility, and work tenure groups (all $p < 0.05$; Table 1).

### Analysis of the relationship between psychological capital and demographic characteristics

The scores for the self-efficacy, hope, resilience, and optimism components of psychological capital were 26.87 ± 5.70, 26.47 ± 5.74, 26.97 ± 5.55, and 26.86 ± 5.59, respectively. Table 2 shows the results of a multiple linear regression analysis to determine the factors associated with the psychological capital of GPs. The psychological capital of GPs was set as the dependent variable, while the other factors were independent variables. The psychological capital scores of GPs were associated with age (β = 0.214, p = 0.003), marital status (β = -3.084, p = 0.012), income level (β = 3.108, p = 0.000), and overtime work (β = -3.399, p = 0.000).

### Analysis of the relationship between patient's contempt and demographic characteristics

The score for patient's contempt was 34.19 ± 7.59. Table 3 shows the results of a multiple linear regression analysis to determine the factors associated with patient's contempt of GPs. Patient's contempt of GPs was set as the dependent variable, while the other factors were independent variables. Factors that were associated with patient's contempt of GPs were age

**Table 1. Professional identity scores by different demographic characteristics.**

| Variables | N (%) | Professional identity score | F/t | P |
|---|---|---|---|---|
| **Age group (years)** | | | 26.672 | 0.001 |
| <30 | 465 (21.30) | 51.63 ± 6.78 | | |
| 30~ | 735 (33.70) | 53.28 ± 6.33 | | |
| 40~ | 728 (33.40) | 54.66 ± 6.07 | | |
| >50 | 252 (11.60) | 54.99 ± 6.02 | | |
| **Sex** | | | 2.917 | 0.004 |
| Men | 1009 (46.28) | 54.02 ± 6.47 | | |
| Woman | 1171 (53.72) | 53.22 ± 6.36 | | |
| **Education level** | | | 3.601 | 0.001 |
| Associate's degree or vocational diploma | 740 (33.94) | 54.28 ± 6.11 | | |
| Bachelor's degree or higher | 1416 (66.06) | 53.24 ± 6.55 | | |
| **Marital status** | | | 5.263 | 0.001 |
| Married | 1764 (80.92) | 53.94 ± 6.27 | | |
| Unmarried/widow/divorced | 416 (19.08) | 52.11 ± 6.82 | | |
| **Contract status** | | | -0.819 | 0.413 |
| Permanent | 1889 (86.65) | 53.55 ± 6.46 | | |
| Temporary | 291 (13.35) | 53.88 ± 6.19 | | |
| **Average monthly income (CNY)** | | | 29.37 | 0.001 |
| <5000 | 1096 (50.28) | 52.64 ± 6.51 | | |
| 5000–8000 | 965 (44.27) | 54.35 ± 6.16 | | |
| >8000 | 119 (5.46) | 56.20 ± 6.20 | | |
| **Professional title** | | | 8.215 | 0.001 |
| Elementary or below | 1157 (53.07) | 53.19 ± 6.54 | | |
| Intermediate | 743 (34.08) | 53.72 ± 6.24 | | |
| Senior | 280 (12.84) | 54.89 ± 6.22 | | |
| **Management responsibility** | | | 6.716 | 0.001 |
| Yes | 551 (25.28) | 55.16 ± 6.23 | | |
| No | 1629 (74.72) | 53.06 ± 6.40 | | |
| **Working overtime** | | | 0.531 | 0.588 |
| Never | 53 (2.43) | 54.25 ± 6.70 | | |
| Occasionally | 943 (43.26) | 53.68 ± 5.99 | | |
| Frequently | 1184 (54.31) | 53.49 ± 6.73 | | |
| **Work tenure (years)** | | | 20.374 | 0.001 |
| <5 | 580 (26.61) | 52.51 ± 6.85 | | |
| 5– | 537 (24.63) | 52.85 ± 6.32 | | |
| 10– | 381 (17.48) | 53.62 ± 6.20 | | |
| 15~ | 682 (31.28) | 55.07 ± 5.96 | | |
| **Practice setting** | | | -0.540 | 0.589 |
| Community health center | 829 (38.03) | 53.50 ± 6.49 | | |
| Township health center | 1351 (61.97) | 53.65 ± 6.38 | | |
| **Too many home visits and too much data-sorting work** | | | 0.641 | 0.521 |
| Yes | 1565 (71.79) | 53.65 ± 6.50 | | |
| No | 615 (28.21) | 53.45 ± 6.21 | | |

($p < 0.05$) ($\beta$ = -0.115, p = 0.000), sex ($\beta$ = -1.570, p = 0.000), professional title ($\beta$ = 1.075, p = 0.000), management responsibility ($\beta$ = 1.102, p = 0.000), and overtime work ($\beta$ = 1.212, p = 0.000).

**Table 2. Multiple linear regression between psychological capital and demographic characteristics.**

| Variables | β | SE | T | P |
|---|---|---|---|---|
| Age (years) | 0.214 | 0.072 | 2.969 | 0.003 |
| Sex | -1.427 | 0.942 | -1.516 | 0.130 |
| Education level | -0.419 | 1.049 | -0.399 | 0.690 |
| Marital status | -3.084 | 1.220 | -2.528 | 0.012 |
| Contract status | 2.646 | 1.357 | 1.950 | 0.051 |
| Average monthly income | 3.108 | 0.809 | 3.841 | 0.000 |
| Professional title | -0.373 | 0.774 | -0.483 | 0.629 |
| Management responsibility | -0.409 | 1.102 | -0.371 | 0.710 |
| Overtime work | -3.399 | 0.828 | -4.104 | 0.000 |
| Work tenure (years) | 0.916 | 0.481 | 1.904 | 0.057 |
| Practice setting | 1.829 | 0.956 | 1.913 | 0.056 |
| Too many home visits and too much data-sorting work | 0.339 | 1.011 | 0.335 | 0.737 |

*The control group consisted of age<30, male, associate's degree or vocational diploma, married, permanent worker, average monthly income<5000, elementary education or below, management responsibility (yes), never work overtime, work tenure<5 years, community health center, and too many home visits and too much data-sorting work (yes).

## Correlation analysis of psychological capital, perceived patient's contempt, and professional identity

Table 4 shows the correlation coefficients between professional identity, psychological capital, and perceived patient's contempt. Specifically, the professional identity of GPs was positively correlated with the four dimensions of psychological capital and negatively correlated with patient's contempt (all $p < 0.01$).

## Multiple linear regression of the professional identity of GPs

Table 5 shows the results of a multiple linear regression analysis to determine factors associated with the professional identity of GPs. The GP professional identity score was set as the

**Table 3. Multiple linear regression between patient's contempt and demographic characteristics.**

| Variables | β | SE | T | P |
|---|---|---|---|---|
| Age (years) | -0.115 | 0.027 | -4.294 | 0.000 |
| Sex | -1.570 | 0.351 | -4.468 | 0.000 |
| Education level | 0.154 | 0.391 | 0.394 | 0.693 |
| Marital status | -0.152 | 0.455 | -0.335 | 0.738 |
| Contract status | -0.361 | 0.506 | -0.712 | 0.476 |
| Average monthly income | -0.226 | 0.302 | -0.749 | 0.454 |
| Professional title | 1.075 | 0.289 | 3.723 | 0.000 |
| Management responsibility | 1.102 | 0.411 | 2.680 | 0.007 |
| Overtime work | 1.212 | 0.309 | 3.919 | 0.000 |
| Work tenure (years) | 0.025 | 0.179 | 0.137 | 0.891 |
| Practice setting | 0.178 | 0.357 | 0.499 | 0.618 |
| Too many home visits and too much data-sorting work | -0.417 | 0.377 | -1.105 | 0.269 |

*The control group consisted of age<30, male, associate's degree or vocational diploma, married, permanent worker, average monthly income<5000, elementary education or below, management responsibility (yes), never work overtime, work tenure<5 years, community health center, and too many home visits and too much data-sorting work (yes).

**Table 4. Correlation analysis of psychological capital and professional identity.**

|  | Professional identity | Self-efficacy | Hope | Resilience | Optimism | Patient's contempt |
|---|---|---|---|---|---|---|
| **Professional identity** | 1 | - | - | - | - | - |
| **Self-efficacy** | 0.257** | 1 | - | - | - | - |
| **Hope** | 0.405** | 0.746** | 1 | - | - | - |
| **Resilience** | 0.393** | 0.670** | 0.867** | 1 | - | - |
| **Optimism** | 0.430** | 0.630** | 0.822** | 0.863** | 1 | - |
| **Patient's contempt** | -0.138** | 0.056** | 0.001 | 0.007 | 0-.014 | 1 |

Note

** $p < 0.001$.

dependent variable, while the other factors were independent variables. The factors associated with the GPs' professional identity score were education level (β = -0.720, p = 0.014), income level (β = 1.018, p = 0.000), management responsibility (β = -1.456, p = 0.000), work tenure (β = 0.440, p = 0.001), self-efficacy (β = -0.122, p = 0.000), hope (β = 0.249, p = 0.000), optimism (β = 0.333, p = 0.000), and patient's contempt (β = -0.103, p = 0.000) (p < 0.05).

## Discussion

This study, which was conducted in the context of the normalization of COVID-19, was the first to investigate the status and factors associated with GPs' professional identity, psychological capital, and patient's contempt in Chongqing, China. It shows that the professional identity of GPs in Chongqing is high, but the mental health of GPs and the current doctor–patient

**Table 5. Multiple linear regression of GP's professional identity.**

| Variables | β | SE | T | P |
|---|---|---|---|---|
| **Age (years)*** | -0.013 | 0.020 | -0.656 | 0.512 |
| **Sex*** | 0.038 | 0.265 | 0.145 | 0.885 |
| **Education level*** | -0.720 | 0.293 | -2.461 | 0.014 |
| **Marital status*** | -0.393 | 0.341 | -1.152 | 0.249 |
| **Contract status*** | 0.345 | 0.379 | 0.909 | 0.363 |
| **Average monthly income*** | 1.018 | 0.227 | 4.493 | 0.000 |
| **Professional title*** | -0.054 | 0.217 | -0.248 | 0.804 |
| **Management responsibility*** | -1.456 | 0.308 | -4.728 | 0.000 |
| **Overtime work*** | 0.138 | 0.233 | 0.591 | 0.555 |
| **Work tenure (years)*** | 0.440 | 0.134 | 3.275 | 0.001 |
| **Practice setting*** | -0.004 | 0.267 | -0.014 | 0.989 |
| **Too many home visits and too much data-sorting work*** | 0.045 | 0.282 | 0.158 | 0.874 |
| **Self-efficacy** | -0.122 | 0.032 | -3.809 | 0.000 |
| **Hope** | 0.249 | 0.049 | 5.115 | 0.000 |
| **Resilience** | -0.003 | 0.051 | -0.064 | 0.949 |
| **Optimism** | 0.333 | 0.045 | 7.437 | 0.000 |
| **Patient's contempt** | -0.103 | 0.016 | -6.376 | 0.000 |

*The control group was age<30, male, associate's degree or vocational diploma, married, permanent worker, average monthly income<5000, elementary education or below, management responsibility (yes), never work overtime, work tenure<5 years, community health center, and too many home visits and too much data-sorting work (yes).

environment continue to be defined by urgent issues that need attention. The professional identity of GPs is related to psychological capital and patients' contempt.

Our results showed that the professional identity of GPs was proportional to their income level; the latter result is consistent with the findings of prior studies. Most GPs in China serve at the baseline level of the healthcare system and tend to have heavier workloads and lower incomes than do medical specialists. Previous studies showed that a high income can help GPs overcome life difficulties, ensure self-worth, and indirectly increase their professional identity [34–36].

The professional identity of GPs with more work experience is stronger than that of GPs with fewer years of experience; increased years working significantly promotes professional identity. When young doctors are expected to skillfully solve clinical problems early in their clinical career, they are prone to psychological pain, which has a negative impact on the formation of professional identity. In other words, because young doctors are new to clinical work, they have less medical experience than older doctors, leading young doctors to feel that their medical knowledge is poor compared with their counterparts [37, 38]. This may also explain why the degree of professional identity in the early career stage was generally low. As work tenure increases, GPs gain a deeper understanding of their occupation, increase their performance quality, and obtain a greater sense of achievement in their work, which may reduce job burnout and improve their enthusiasm for their work; in turn, these changes may enable them to feel a stronger sense of professional identity.

GPs who held administrative roles had a stronger sense of professional identity than those who did not hold such roles. This may be because GPs in administrative positions have more opportunities for career development, which may contribute to their job satisfaction and sense of personal value [39]. Meanwhile, the more developed the GP's educational background, the poorer their sense of professional identity. GPs with higher education are subject to higher social expectations; thus, they hope to obtain corresponding social status, but the social recognition of GPs in China is generally poor. If these observations are accurate, they require more work from interested parties at the grassroots level to be addressed. Based on these findings, managers should adopt corresponding measures and strategies to stabilize the professional identity of GPs. For example, they could do so by increasing GP income, regularly checking GP mental health, and creating a harmonious doctor–patient environment.

This study also found that age, marital status, average monthly income, and overtime work were related to psychological capital. As they age, GPs gradually transition from students to practitioners. During this shift, they increase their connections with patients in their daily work, often become more flexible in dealing with different issues, and gradually become more mature.

Regarding marital status, being married was shown to strengthen psychological capital; this may be because married GPs may be able to enjoy timely support from their partners and families when they encounter unpleasant situations in life [25]. This may, in turn, reduce their incidence of depression and strengthen their psychological capital.

Moreover, an increase in income was also shown to promote psychological capital. In China, it is generally considered that GPs do not have as high status as specialist physicians, leading to the low social status and income of the former. Furthermore, because the growth cycle for this medical career is relatively long, GPs need to work and conduct research to accelerate their promotions [40]. Nevertheless, GPs who work in relatively large hospitals have access to fewer resources, which can make their road to promotion more difficult. When this is coupled with the effort that early-career GPs tend to put forth, fatigue can occur, and their efforts can become mismatched with their returns. This reality may also lead GPs with lower psychological capital to have lower income [41].

Additionally, GPs in our sample who frequently worked overtime were associated with lower levels of psychological capital [42]. Compared with medical specialists, who tend to focus on the treatment of specific diseases, GPs' daily work requires them to think about several different patient needs in a high-pressure work environment. Considering these aspects of their work, GPs may be likely to experience mood disorders or symptoms, such as depressive episodes or anxiety. If these are not addressed appropriately and in a timely manner, GPs' psychological capital may decline. Internet cognitive behavioral therapy may be an effective way to solve this problem [43]. During the pandemic, a study in Singapore showed that an online cognitive behavioral therapy intervention could enhance people's ability to manage stress, combat anxiety, and prevent depression, thus improving the psychological capital of GPs [44, 45].

Many previous studies have shown that higher psychological capital can play a positive role in how GPs treat their patients during a consultation, significantly improve negative job-related emotions, and enable more proactive behaviors. Since GPs must engage a large number of patients every day, they inevitably encounter patients who may not like them or treat them kindly. GPs with stronger psychological structures may be able to recover from such situations more quickly than others and may also be more likely to deal with such situations without experiencing depression or depressive symptoms [46].

A good doctor–patient relationship can improve patients' satisfaction and help them better adhere to the treatment goals and trust the advice from doctors; furthermore, such a relationship can reduce the occurrence of medical errors by doctors. However, in China, the current situation regarding patients' distrust of physicians has become so problematic that physician–patient conflicts often emerge; this, in turn, leads to an increase in patients' doubts regarding their GPs' professionalism and skills, which can negatively impact patient prognosis. Many factors affect the doctor–patient relationship, such as the mismatch of professional knowledge between doctors and patients, which is difficult to change. Moreover, the workload of GPs in China is heavy; thus, the time allocated to each patient is relatively short. Doctors may spend more time establishing medical history and treatment, paying less attention to the situation other than the disease. Thus, the patient's experience of seeing a doctor is ignored, which leads to a decrease in the patient's recognition of the doctor, thus affecting the doctor–patient relationship [47, 48]. By contrast, although China's medical technology is comparable with that of the rest of the world, owing to the relatively late development of general practice, the basic medical insurance system continues to be a "disease-centered" structural framework, despite the introduction of medical reforms 10 years ago; therefore, it remains difficult to achieve "bio-psycho-social" patient-centered medicine. Moreover, the working environment of GPs is difficult to compare with that in large hospitals. Many examinations are difficult to perform, and even drugs with better curative effects for basic diseases are difficult to obtain, such that they have to be replaced by other drugs with the same effects. In the fight against the COVID-19 pandemic, Chinese medical workers have shown their responsibility, and several reports have hailed them as heroes. This has improved the relationship between doctors and patients, but has not been sufficient to change the current situation [49–51]. While GPs often attempt to solve their patients' problems, their patients may perceive their treatments as unsatisfactory, leading GPs to doubt own their own skills and abilities and have poor professional identity [52]. In particular, in poor long-term physician–patient relationships, GPs' positive feelings regarding their profession may diminish, decreasing their professional identity while increasing doubt in their professional skills and their turnover rate. Our results are not entirely concordant with prior evidence, as self-efficacy negatively influenced professional identity among our sample of GPs [53, 54]. This may be related to the Chinese setting of the study; specifically, Chinese GPs may be confident in their practical abilities, but the physician–patient

relationships they encounter may be characterized by a lack of trust. Patients often have doubts about doctors' treatment and feel that doctors do not try their best to solve problems for themselves, which leads to an unsatisfactory curative effect in the treatment process. This may lead doctors to experience long periods of doubt in their professional ability and a decline of their professional identity.

This study also shows that higher hope and optimism are associated with improved GP professional identity. Thus, positive psychology may help GPs adjust to the pressures they experience during clinical practice, increase investment in work, and positively affect their performance, potentially promoting psychological well-being in daily work and job re-engagement [55].

Furthermore, the survey results showed that GPs with higher patient's contempt scores had lower professional identity. Before the new health care reform in China was implemented, some widespread phenomena (e.g., doctors being given rebates by prescribing designated drugs to patients, patients giving doctors monetary gifts) resulted in increasingly tense doctor–patient relationships and progressively increased patient distrust [56, 57]. In other countries, GPs are relatively older and more experienced, which can usually better help them solve patients' medical problems. However, in China, owing to the late establishment of the GP system and its imperfections, most GPs are young people who have recently graduated from medical colleges. https://fanyi.baidu.com//?channel=pcPinzhuan - ##https://fanyi.baidu.com//?channel=pcPinzhuan - ##javascript:void(0);Compared with older GPs, they have little clinical knowledge; this can cause them to have a poorer ability to carry out professional processes, which indirectly increases the unfriendly relationship between doctors and patients. The physician–patient relationship is a special interpersonal relationship, and good communication can improve it and enable patients to shift from passively to actively engaging in their treatment. Through patients' active participation, GPs can better understand their actual needs, solve problems from the patient's perspective, obtain patient understanding and trust, and gradually decrease patient's contempt regarding medical care and practitioners [58, 59]. In such scenarios, the physician–patient relationship becomes more cooperative; moreover, when the goals and efforts of both GPs and patients become more consistent, the physician–patient relationship may improve, and physicians' perceptions of patient's contempt may diminish. Promoting cooperation in physician–patient relationships may also help GPs more clearly recognize the value of their profession and improve their professional identity.

## Limitations

This study has several limitations. First, it adopted a cross-sectional design; therefore, it was not able to determine causal relationships between the variables of interest. Second, other potential predictors of GP professional identity (e.g., work stress, workplace violence, team support) were not included in our questionnaire, and future researchers should further explore the influence of these factors on professional identity. Third, our sample was from a municipality in western China; thus, it may not be applicable to the situation of GPs across the country; future studies should use larger sample sizes.

## Conclusions

The normalization of the COVID-19 pandemic has brought great challenges to China's primary health system, in which GPs are an indispensable element. GPs' sense of professional identity affects their daily work efficiency and stability. This study found that income level, position type (administrative/non-administrative), work duration, self-efficacy, hope, optimism, and contempt for patients affect professional identity among GPs in China. If managers

fail to pay attention to these factors in a timely manner, GP talent may be lost. These survey results can provide a basis for relevant departments to formulate specific strategies for GPs. At the individual level, GPs should strengthen their mental health by developing a positive and optimistic attitude toward difficulties. Efforts should be made at the national level to build a friendly doctor–patient environment. Notably, these interventions could improve general practice and reduce brain drain.

## Supporting information

**S1 File.**
(DOCX)

**S1 Dataset.**
(XLSX)

## Acknowledgments

We want to thank Chongqing Health Commission and Science and Technology Bureau for support in data acquisition.

## Author Contributions

**Formal analysis:** Qiaoya Li.

**Investigation:** Jingzhi Deng.

**Methodology:** Yang Xu, Qiaoya Li, Wen Yang.

**Supervision:** Huisheng Deng.

**Validation:** Jingzhi Deng.

**Writing – original draft:** Jingzhi Deng.

**Writing – review & editing:** Jingzhi Deng, Yang Xu.

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
