## [Decision Letter · Decision Letter 0]

28 Feb 2023

PONE-D-22-31263The relationship between psychological capital,patient's contempt and professional identity among general practitioners in the era of COVID-19 in Chongqing,ChinaPLOS ONE

Dear Dr. Deng,

Thank you for submitting your manuscript to PLOS ONE. After careful consideration, we feel that it has merit but does not fully meet PLOS ONE’s publication criteria as it currently stands. Therefore, we invite you to submit a revised version of the manuscript that addresses the points raised during the review process.

We look forward to receiving your revised manuscript.

Kind regards,

Vanessa Carels

Staff Editor

PLOS ONE

Journal Requirements:

Reviewers' comments:

Reviewer's Responses to Questions

**Comments to the Author**

1. Is the manuscript technically sound, and do the data support the conclusions?

Reviewer #1: Yes

Reviewer #2: Yes

2. Has the statistical analysis been performed appropriately and rigorously? 

Reviewer #1: Yes

Reviewer #2: Yes

3. Have the authors made all data underlying the findings in their manuscript fully available?

Reviewer #1: Yes

Reviewer #2: Yes

4. Is the manuscript presented in an intelligible fashion and written in standard English?

Reviewer #1: Yes

Reviewer #2: Yes

5. Review Comments to the Author

Reviewer #1: I congratulate you for this paper in a field that is also becoming important in other parts of the globe.

Please find attached my bullets to help increase the qualoity of your onservational study. We can only know about something once we have pictured it. Then intervention can be made after a very sound thought of what to do. Allways think about yhe "Mikado" game.

Reviewer #2: I found this study quite interesting. Congratulations to the authors. I just have a very small suggestion.

Introduction

“Self-efficacy refers to one’s confidence in own competency for completing a task, facing challenges, and striving for success. Optimism refers to when a person has a positive attribution style and 90 attitude towards both the present and the future. Resilience is defined as the ability to quickly recover from adversity, setbacks, and failures, and even positively transform and grow after such happenings. Hope is a state of positive motivation to achieve a predetermined goal through various means[22].”

This paragraph breaks the integrity of the introduction. My suggestion: 94.-101. It can be added to the end of the paragraph in the page range.

Methods

Cronbach's alpha values of the scales should be given.

6. PLOS authors have the option to publish the peer review history of their article (what does this mean?). If published, this will include your full peer review and any attached files.

Reviewer #1: **Yes: **Luiz Miguel Santiago

Reviewer #2: No

---

## [Author Response · Author response to Decision Letter 0]

13 Mar 2023

Response to the Editor’s note and suggestions

We have uploaded the minimal anonymized data as Supporting Information files. Thank you for your advice.

Response to Reviewer 1

Comment 1:A reference on the type of choice of population must be met here: convenience? random?

Response 1: Thank you for your advice. We have added to the sentence an explanation of how the population was chosen; the population was selected randomly.

Comment 2:The specific scales used must be mentioned in the abstract.

Response 2: Thank you for your constructive comments. We have addressed the problem you mentioned. The specific scales used are now stated in the abstract, which has been modified as follows: "General practitioners’ sense of professional identity, mental health, and sense of patients' disrespect were measured using the Professional Identity Scale, Psychological Capital Questionnaire, and Patient's Contempt Questionnaire. Sociodemographic characteristics were also collected.” (Page 2, lines 32-35)

Comment 3: Validity of all scales used, namely the Cronbach's alfa shoulde be refered.

Response 3: Thank you for your suggestion. We have calculated Cronbach's alpha of all scales in the Measures section and added them to the end of the respective sentences. The Cronbach's alpha values of the Professional Identity Scale, Psychological Capital Questionnaire, and Patient's Contempt Questionnaire were 0.936, 0.977, and 0.914, respectively.

Comment 4:responsibility... butthis sentende must be corrected.

Response 4: Thank you for your comments. We also see the problem with this sentence. Therefore, we have revised to "General practitioners with higher education are subject to higher social expectations; thus, they hope to obtain corresponding social status. " (Page 19, lines 310-313)

Comment 5:The second GPS must be altered to GPs. Could a mention about the patiente-doctor relationship be included? How come this is about not using Patient Centered Medicine? How come this is due to over use of medicines and under expeted use of analysis and othe neans of study like radiographs and other exmas? What about the media's press on patients? 

Response 5: Thank you for your opinion, which is very helpful in the improvement of our article. We have revised the second GPS to GPs. Regarding your reference to the sentence "While GPs often try to solve their patients' problems, their patients may perceive their treatments as unsatisfactory, leading GPs to doubts about own their own skills and abilities with poor professional identity," you asked whether this sentence can be considered from other perspectives. After consideration, we think that doing so is highly necessary; thus, we read relevant references and added a supplementary explanation of this sentence, hoping to meet your requirements. (Pages 21-22, lines 357-377)

References

44. de Waard CS, Poot AJ, den Elzen WPJ, Wind AW, Caljouw MAA, Gussekloo J. Perceived doctor-patient relationship and satisfaction with general practitioner care in older persons in residential homes. Scand J Prim Health Care. 2018;36(2):189-97. doi: 10.1080/02813432.2018.1459229. PubMed PMID: 29644911.

45. Zhong C, Zhou M, Luo Z, Liang C, Li L, Kuang L. Association between doctor-patient familiarity and patient-centred care during general practitioner's consultations: a direct observational study in Chinese primary care practice. BMC Fam Pract. 2021;22(1):107. doi: 10.1186/s12875-021-01446-4. PubMed PMID: 34049489.

46. Zhou Y, Ma Y, Yang WFZ, Wu Q, Wang Q, Wang D, et al. Doctor-patient relationship improved during COVID-19 pandemic, but weakness remains. BMC Fam Pract. 2021;22(1):255. doi: 10.1186/s12875-021-01600-y. PubMed PMID: 34937550.

47. Zhou Y, Yang WFZ, Ma Y, Wu Q, Yang D, Liu T, et al. Doctor-Patient Relationship in the Eyes of Medical Professionals in China During the COVID-19 Pandemic: A Cross-Sectional Study. Front Psychiatry. 2021;12:768089. doi: 10.3389/fpsyt.2021.768089. PubMed PMID: 34777069.

48. Zhou Y, Chen S, Liao Y, Wu Q, Ma Y, Wang D, et al. General Perception of Doctor-Patient Relationship From Patients During the COVID-19 Pandemic in China: A Cross-Sectional Study. Front Public Health. 2021;9:646486. doi: 10.3389/fpubh.2021.646486. PubMed PMID: 34295863.

Comment 6:I do not think you have a limitation here! What other ways do you have to undestand a problem? A prospective study would mean a first picture which is what you have done. We can only see a problem when we picture it. Then we can intervene. 

Response 6: Thank you for your constructive comments. We have revised the limitations according to your suggestion.

Response to Reviewer 2

Comment 1:Introduction:“Self-efficacy refers to one’s confidence in own competency for completing a task, facing challenges, and striving for success. Optimism refers to when a person has a positive attribution style and 90 attitude towards both the present and the future. Resilience is defined as the ability to quickly recover from adversity, setbacks, and failures, and even positively transform and grow after such happenings. Hope is a state of positive motivation to achieve a predetermined goal through various means[22].”This paragraph breaks the integrity of the introduction. My suggestion: 94.-101. It can be added to the end of the paragraph in the page range.

Response 1: Thank you for your constructive comments. We noticed that the question you raised is very helpful in improving the quality of our article. Therefore, we have adjusted the word order of this portion of the sentence to address your concerns.

Comment 2:Methods:Cronbach’s alpha values of the scales should be given.

Response 2: Thank you for your advice. We have calculated the Cronbach's alpha of all scales in the Measures section and added them to the end of the respective sentences. The Cronbach's alpha values of the Professional Identity Scale, Psychological Capital Questionnaire, and Patient's Contempt Questionnaire were 0.936, 0.977, and 0.914, respectively.

---

## [Decision Letter · Decision Letter 1]

18 Apr 2023

PONE-D-22-31263R1The relationship between psychological capital,patient's contempt and professional identity among general practitioners in the era of COVID-19 in Chongqing,ChinaPLOS ONE

Dear Dr. Deng,

Thank you for submitting your manuscript to PLOS ONE. After careful consideration, we feel that it has merit but does not fully meet PLOS ONE’s publication criteria as it currently stands. Therefore, we invite you to submit a revised version of the manuscript that addresses the points raised during the review process.

Dear author This is an interesting paper, but final revisions are needed for publication. This manuscript does not mention burnout, so there is no need to modify the burnout mentioned by reviewer 3. Please revise other issues.

We look forward to receiving your revised manuscript.

Kind regards,

Chunyu Zhang

Academic Editor

PLOS ONE

Additional Editor Comments:

Dear author

This is an interesting paper, but final revisions are needed for publication. I hope the following comments are helpful to you.

1. There are no examples in your measurement tool.

2. Lack of research hypotheses.

3. The author adopted a sampling method (stratified random sampling) with probability, but the expression in the paper does not show probability.

4. Lack of convergence validity and discriminative validity tests.

5. Please refer to other papers and revised the presentation of Table 4, which is inconsistent with the common correlation analysis.

6. Lack of CMV analysis.

7. Lines 295-299, "A potential reason for this is that physicians have longer working hours than those in other occupations; this may cause them to experience more job-related pressures early on in their careers, which can lead to job burnout" This sentence is very farfetched in explaining the previous sentence.

8. Please proofread the grammar.

Reviewers' comments:

Reviewer's Responses to Questions

**Comments to the Author**

1. If the authors have adequately addressed your comments raised in a previous round of review and you feel that this manuscript is now acceptable for publication, you may indicate that here to bypass the “Comments to the Author” section, enter your conflict of interest statement in the “Confidential to Editor” section, and submit your "Accept" recommendation.

Reviewer #2: All comments have been addressed

Reviewer #3: (No Response)

2. Is the manuscript technically sound, and do the data support the conclusions?

Reviewer #2: Yes

Reviewer #3: Yes

3. Has the statistical analysis been performed appropriately and rigorously? 

Reviewer #2: Yes

Reviewer #3: Yes

4. Have the authors made all data underlying the findings in their manuscript fully available?

Reviewer #2: Yes

Reviewer #3: Yes

5. Is the manuscript presented in an intelligible fashion and written in standard English?

Reviewer #2: Yes

Reviewer #3: Yes

6. Review Comments to the Author

Reviewer #2: (No Response)

Reviewer #3: I have the following comments for the authors to address and I am happy to review this paper again:

1) This study did not mention about burnout. Please state the concept of COVID-19 burnout and burnout among physicians:

Search PubMed for: Burnout is an important public health issue at times of the COVID-19 pandemic. Current measures which focus on work-based burnout have limitations in length and/or relevance. When stepping into the post-pandemic as a new Norm Era, the burnout scale for the general population is urgently needed to fill the gap. This study aimed to develop a COVID-19 Burnout Views Scale (COVID-19 BVS) to measure burnout views of the general public in a Chinese context and examine its psychometric properties.

Search PubMed for: Our findings suggest a high prevalence of burnout among medical and surgical residents. Older and male residents suffered more than their respective counterparts.

2) The authors stated "GPs may be likely to experience mood disorders or symptoms, such as

354 depressive episodes or anxiety. If these are not addressed appropriately and in a timely

355 manner, then GPs’ psychological capital may decline". Please discuss internet psychological interventions to help GPs based on the following:

The most evidence-based treatment is cognitive behaviour therapy (CBT), especially Internet CBT that can prevent the spread of infection during the pandemic.

Use of Cognitive Behavior Therapy (CBT) to treat psychiatric symptoms during COVID-19:

Mental Health Strategies to Combat the Psychological Impact of COVID-19 Beyond Paranoia and Panic. Ann Acad Med Singapore. 2020;49(3):155‐160.

Cost-effectiveness of iCBT:

Moodle: The cost effective solution for internet cognitive behavioral therapy (I-CBT) interventions. Technol Health Care. 2017;25(1):163-165. doi: 10.3233/THC-161261. PMID: 27689560.

Internet CBT can treat psychiatric symptoms such as insomnia:

Efficacy of digital cognitive behavioural therapy for insomnia: a meta-analysis of randomised controlled trials. Sleep Med. 2020 Aug 26;75:315-325. doi: 10.1016/j.sleep.2020.08.020. Epub ahead of print. PMID: 32950013.

7. PLOS authors have the option to publish the peer review history of their article (what does this mean?). If published, this will include your full peer review and any attached files.

Reviewer #2: No

Reviewer #3: No

---

## [Author Response · Author response to Decision Letter 1]

9 May 2023

Response to the Editor and Reviewers

We thank both the editor and the reviewers for their positive and constructive comments and suggestions.

Response to the Editor’s note and suggestions

Comment 1. There are no examples in your measurement tool.

Response 1: Thank you for your comment. We have added examples of the measurement tool (lines 160–185).

Comment 2. Lack of research hypotheses.

Response 2: Thank you for your comment. We hypothesized that the professional identity of general practitioners would be improved during COVID-19, and greater psychological capital would promote the professional identity of these persons, while patient’s contempt might lead to a reduction in the professional identity of general practitioners. We have added this sentence to the article (lines 125–128).

Comment 3. The author adopted a sampling method (stratified random sampling) with probability, but the expression in the paper does not show probability.

Response 3: Thank you for your comment. We apologize for our lack of clarity. We adopted a randomized cluster sampling method, in which we selected 23 districts and counties; we recruited all general practitioners who met the inclusion and exclusion criteria in the selected areas(lines 133-143).

Comment 4. Lack of convergence validity and discriminative validity tests.

Response 4: Thank you for noting these issues. The AVE, CR and sqrt(AVE) value of the Professional Identity Scale are 0.450,0.910,0.671. The Psychological Capital Questionnaire includes four dimensions: self-efficacy, hope, resilience and optimism. The AVE, CR, and sqrt(AVE) values are 0.596,0.897,0.772 in the self-efficacy dimension; 0.650,0.918,0.806 in the hope dimension; 0.682,0.927,0.826 in the resilience dimension; 0.711,0.937,0.843 in the optimism dimension.The AVE, CR and sqrt(AVE) value of the Patient’s Contempt Questionnaire are0.543,0.922,0.737.The convergence validity and discriminative validity of the questionnaire are reliable.

Comment 5. Please refer to other papers and revised the presentation of Table 4, which is inconsistent with the common correlation analysis.

Response 5: Thank you for your constructive comment. We have read other articles and modified Table 4 accordingly, which improved the presentation of our article.

Comment 6. Lack of CMV analysis.

Response 6: Thank you for your advice. We conducted CMV analysis of the questionnaire data via Harman's single-factor test. The exploratory factor analysis results for the 47 items revealed 7 factors with an eigenvalue greater than 1. The variance explained by the first factor was 34.148% (<50%), indicating no serious common method bias in this study.

Comment 7. Lines 295-299, "A potential reason for this is that physicians have longer working hours than those in other occupations; this may cause them to experience more job-related pressures early on in their careers, which can lead to job burnout" .This sentence is very farfetched in explaining the previous sentence.

Response 7: Thank you for your constructive comment. After reading the relevant literature, we changed this sentence to " The professional identity of GPs with more work experience is stronger than that of GPs with fewer years of experience; increased years working significantly promotes professional identity. When young doctors are expected to skillfully solve clinical problems early in their clinical career, they are prone to psychological pain, which has a negative impact on the formation of professional identity. In other words, because young doctors are new to clinical work, they have less medical experience than older doctors, leading young doctors to feel that their medical knowledge is poor compared with their counterparts." (lines 290–296). I hope this explanation meets your requirements.

References

1.Cohen MJM, Kay A, Youakim JM, Balaicuis JM. Identity transformation in medical students. Am J Psychoanal. 2009;69(1):43-52. doi: 10.1057/ajp.2008.38. PubMed PMID: 19295620.

2.Bynum WE, Artino AR, Uijtdehaage S, Webb AMB, Varpio L. Sentinel Emotional Events: The Nature, Triggers, and Effects of Shame Experiences in Medical Residents. Acad Med. 2019;94(1):85-93. doi: 10.1097/ACM.0000000000002479. PubMed PMID: 30277959.

Comment 8. Please proofread the grammar.

Response 8:Thank you for your constructive comment. We have proofread the grammar and made corrections as needed.

Response to Reviewer 3

Comment 1:This study did not mention about burnout. Please state the concept of COVID-19 burnout and burnout among physicians:

Search PubMed for: Burnout is an important public health issue at times of the COVID-19 pandemic. Current measures which focus on work-based burnout have limitations in length and/or relevance. When stepping into the post-pandemic as a new Norm Era, the burnout scale for the general population is urgently needed to fill the gap. This study aimed to develop a COVID-19 Burnout Views Scale (COVID-19 BVS) to measure burnout views of the general public in a Chinese context and examine its psychometric properties.

Search PubMed for: Our findings suggest a high prevalence of burnout among medical and surgical residents. Older and male residents suffered more than their respective counterparts.

Response 1: The journal editor stated "This manuscript does not mention burnout, so there is no need to modify the burnout mentioned by reviewer 3. Please revise other issues.” Therefore, we did not revise the text to mention burnout.

Comment 2:) The authors stated "GPs may be likely to experience mood disorders or symptoms, such as

354 depressive episodes or anxiety. If these are not addressed appropriately and in a timely

355 manner, then GPs’ psychological capital may decline". Please discuss internet psychological interventions to help GPs based on the following:

The most evidence-based treatment is cognitive behaviour therapy (CBT), especially Internet CBT that can prevent the spread of infection during the pandemic.

Use of Cognitive Behavior Therapy (CBT) to treat psychiatric symptoms during COVID-19:

Mental Health Strategies to Combat the Psychological Impact of COVID-19 Beyond Paranoia and Panic. Ann Acad Med Singapore. 2020;49(3):155‐160.

Cost-effectiveness of iCBT:

Moodle: The cost effective solution for internet cognitive behavioral therapy (I-CBT) interventions. Technol Health Care. 2017;25(1):163-165. doi: 10.3233/THC-161261. PMID: 27689560.

Internet CBT can treat psychiatric symptoms such as insomnia:

Efficacy of digital cognitive behavioural therapy for insomnia: a meta-analysis of randomised controlled trials. Sleep Med. 2020 Aug 26;75:315-325. doi: 10.1016/j.sleep.2020.08.020. Epub ahead of print. PMID: 32950013.

Response 2: Thank you for your comment. We added the following text, accordingly (lines 377–341): "Internet cognitive behavioral therapy may be an effective way to solve this problem. During the pandemic, a study in Singapore showed that an online cognitive behavioral therapy intervention could enhance people’s ability to manage stress, combat anxiety, and prevent depression, thus improving the psychological capital of GPs ." 

References

1.Zhang MWB, Ho RCM. Moodle: The cost effective solution for internet cognitive behavioral therapy (I-CBT) interventions. Technol Health Care. 2017;25(1):163-5. doi: 10.3233/THC-161261. PubMed PMID: 27689560.

2. Soh HL, Ho RC, Ho CS, Tam WW. Efficacy of digital cognitive behavioural therapy for insomnia: a meta-analysis of randomised controlled trials. Sleep Med. 2020;75:315-25. doi: 10.1016/j.sleep.2020.08.020. PubMed PMID: 32950013.

3. Ho CS, Chee CY, Ho RC. Mental Health Strategies to Combat the Psychological Impact of Coronavirus Disease 2019 (COVID-19) Beyond Paranoia and Panic. Ann Acad Med Singap. 2020;49(3):155-60. PubMed PMID: 32200399.

---

## [Decision Letter · Decision Letter 2]

6 Jun 2023

The relationship between psychological capital,patient's contempt, and professional identity among general practitioners during COVID-19 in Chongqing,China

PONE-D-22-31263R2

Dear Dr. Deng,

We’re pleased to inform you that your manuscript has been judged scientifically suitable for publication and will be formally accepted for publication once it meets all outstanding technical requirements.

Kind regards,

Chunyu Zhang

Academic Editor

PLOS ONE

Additional Editor Comments (optional):

Reviewers' comments:

Reviewer's Responses to Questions

**Comments to the Author**

1. If the authors have adequately addressed your comments raised in a previous round of review and you feel that this manuscript is now acceptable for publication, you may indicate that here to bypass the “Comments to the Author” section, enter your conflict of interest statement in the “Confidential to Editor” section, and submit your "Accept" recommendation.

Reviewer #2: All comments have been addressed

Reviewer #3: All comments have been addressed

2. Is the manuscript technically sound, and do the data support the conclusions?

Reviewer #2: Yes

Reviewer #3: Yes

3. Has the statistical analysis been performed appropriately and rigorously? 

Reviewer #2: Yes

Reviewer #3: Yes

4. Have the authors made all data underlying the findings in their manuscript fully available?

Reviewer #2: Yes

Reviewer #3: Yes

5. Is the manuscript presented in an intelligible fashion and written in standard English?

Reviewer #2: Yes

Reviewer #3: Yes

6. Review Comments to the Author

Reviewer #2: (No Response)

Reviewer #3: I recommend publication for The relationship between psychological capital,patient's contempt, and professional

identity among general practitioners during COVID-19 in Chongqing,China

7. PLOS authors have the option to publish the peer review history of their article (what does this mean?). If published, this will include your full peer review and any attached files.

Reviewer #2: No

Reviewer #3: No

---

## [Editor Report · Acceptance letter]

13 Jun 2023

PONE-D-22-31263R2 

The relationship between psychological capital, patient’s contempt, and professional identity among general practitioners during COVID-19 in Chongqing, China 

Dear Dr. Deng:

I'm pleased to inform you that your manuscript has been deemed suitable for publication in PLOS ONE. Congratulations! Your manuscript is now with our production department. 

Kind regards, 

on behalf of

Dr. Chunyu Zhang 

Academic Editor

PLOS ONE